



# Boundary layer structure and parameterization uncertainties affecting wind turbine hub-height predictions: A comparative assessment of New England and Florida

Adam Ayouche[1], Baylor Fox-Kemper[1], and Nathan J M Laxague[2]

[1]Department of Earth, Environmental, and Planetary Sciences (DEEPS), Brown University, Providence, RI, USA
[2]Department of Mechanical Engineering, University of New Hampshire, Durham, NH, USA

**Correspondence:** Adam Ayouche (adam_ayouche@brown.edu)

**Abstract.** As wind energy development expands to diverse geographical regions, understanding the complex atmospheric conditions that turbines encounter is essential for accurate resource assessment and operational forecasting. This study presents an analysis of boundary layer structure, thermodynamic forcing mechanisms, and model parameterization uncertainties affecting hub-height winds using North American Mesoscale Forecast System (NAM) data and one-dimensional RANS simulations.
Our analysis reveals important regional and seasonal variations in atmospheric boundary layer characteristics between New England and Florida. In New England, hub-height to PBL ratios frequently exceed critical thresholds during summer convective conditions, indicating turbine operation above the shallow atmospheric boundary layer, while Florida maintains more consistent conditions with ratios well below these thresholds year-round. Thermodynamic forcing shows similarly distinct patterns, with New England experiencing strong seasonal CAPE variations between winter and summer compared to Florida's consistently elevated values. Systematic evaluation of NAM turbulence parameterization reveals moderate correlations with resolved turbulence but significant biases across the TKE spectrum. Controlled numerical experiments demonstrate that different turbulence closure schemes and vertical resolution configurations produce substantial variations in power density estimates, with the greatest deviations occurring under stable conditions.

## 1 Introduction

Wind energy has emerged as a fundamental resource for global renewable energy strategies, driving targeted future offshore and onshore deployments (International Renewable Energy Agency, 2022; Global Wind Energy Council, 2023). As part of its long-term energy strategy, the United States has set an ambitious goal to reach 30 GW of offshore wind capacity by 2030 (U.S. Department of Energy, 2021). Achieving this requires substantial development of wind farms in coastal regions, a process currently navigating various policy and market complexities, notably influenced by evolving political and federal landscapes, that underscore the need for robust scientific understanding. The continuity of these projects relies on accurate wind forecasting, as power production scales with the cube of wind speed; meaning a 10% error in wind speed translates to approximately 30% error in power estimation (Manwell et al., 2010). This forecasting is particularly challenging over the



span of the turbines, characterized by the hub heights (80-140 m), where complex atmospheric stratification and aerodynamics create unique prediction difficulties (Liu and Stevens, 2021; Kosović et al., 2025).

Modern wind turbines operate at hub heights of 80-140 m, a region within the upper boundary layer or its "cap", i.e., the transition to the free atmosphere (Sanz Rodrigo et al., 2017). This atmospheric zone experiences distinctive wind shear and turbulence intensity that vary significantly as atmospheric stability conditions vary above and below the hub height. Strong solar heating induces convective mixing during summer daytime conditions, while in winter, density stratification (through vertical variations of temperature and humidity) is maintained with weaker vertical mixing. Significant vertical shear and

low-level jets occur during stable nighttime conditions that are frequent year-round and can affect turbine rotors, potentially beneficially. Forecasting based on surface properties, e.g., satellite-sensing of sea surface temperature or roughness, often fails to characterize these flows as they may be decoupled from surface conditions, particularly in stable atmospheric regimes (Weide Luiz and Fiedler, 2022). Other complex phenomena, such as sea breezes, thermal internal boundary layers, and marine atmospheric boundary layer dynamics, also affect the flow at hub height (Talbot et al., 2007).

Numerical weather prediction (NWP) models are crucial for operational wind energy forecasting. Various forecast models serve as a fundamental basis for wind farm layout and implementation, including the North American Mesoscale (NAM, Colbert et al. (2019)) Forecast System model, High-Resolution Rapid Refresh (HRRR, Dowell et al. (2022)), the Global Forecast System (GFS, Yang et al. (2020)), and the European Centre for Medium-Range Weather Forecasts (ECMWF, de Rosnay et al. (2022)) model, each with different spatial resolutions, physics parameterizations, and forecast horizons. Recent

comparisons near one of our regions have shown similar skill at small scales among the GFS, NAM, and HRRR (e.g., Gaudet et al., 2024), but we choose to focus on the NAM because our coastal modeling work Sane et al. (2021) has demonstrated it has high skill in regional offshore winds.

However, boundary layer processes require fine resolutions that cannot be achieved with operational resolutions (typically 3-56km horizontal, and 10 to 250m vertical); this is particularly challenging at hub height, where small vertical displacements can yield large changes in wind speed and direction (Maas, 2023). Most NWP models rely on planetary boundary layer (PBL)

parameterization schemes that often fail to capture the fine-scale vertical structure and temporal evolution of winds at hub height (Siuta et al., 2017; Jia et al., 2023). Therefore, evaluating these models' performance at hub height remains insufficient compared to the numerous studies focused on surface and upper-troposphere validation. This validation gap is especially problematic given that hub-height forecast accuracy directly impacts power production estimates and grid integration strategies

for wind energy resources.

Different atmospheric dynamics in New England and Florida affect potential wind energy resources. New England is characterized by strong seasonal variability with winter northwesterly flows bringing cold, dry continental air masses that create strong thermal gradients offshore, while summer southwesterlies transport warm, moist maritime air (Figure 1). The region hosts the first offshore wind (Block Island Wind Farm) and major offshore wind developments, including Vineyard Wind and

Revolution Wind projects, totaling approximately 5 GW of planned capacity. These developments face complex wind patterns at hub height due to the frequent passage of midlatitude cyclones and anticyclones (Jiang and Perrie, 2007). In contrast, Florida's subtropical environment features more consistent southeasterly trade winds influenced by the Bermuda High, with





variations primarily driven by sea breeze circulation and tropical systems (Bowles and Strazzo, 2024). Though Florida has limited current wind development, its estimated 2 GW offshore potential warrants investigation of forecasting capabilities in

this distinct meteorological regime. Furthermore, despite its current deployment status, Florida serves as an invaluable comparative case study to New England. Its consistently subtropical environment and distinct atmospheric drivers provide a stark meteorological contrast, enabling a robust assessment of forecast model performance across diverse climatological regimes. These regional differences directly impact hub-height flow characteristics and potentially affect NAM and other forecast model performance in ways not yet fully characterized.

Current literature reveals significant knowledge gaps in mesoscale model performance at turbine hub heights offshore. While numerous studies have evaluated atmospheric models for general applications, few have focused on the marine atmospheric boundary layer at the 80-250m heights now critical for offshore wind energy (Shaw et al., 2022). The marine ABL remains poorly observed for wind energy applications, severely limiting model validation and identification of specific physics issues. Current parameterizations often embed assumptions that are routinely violated in offshore environments, particularly regarding

air-sea interaction and stratification effects on wind profiles. The complexity of coastal boundaries generates phenomena such as low-level jets that significantly affect power generation yet remain difficult to simulate accurately (Debnath et al., 2021). Our study addresses the following aspects, all of which relate to the accuracy of present weather forecast models and parameterizations for offshore wind applications: (1) a comprehensive assessment of hub-height location relative to key atmospheric layer scales (PBL height, Obukhov length scale) across the different stability regimes that occur during the diurnal and seasonal

cycles, (2) a systematic evaluation of hub-height dynamics and their response to atmospheric forcings, (3) a detailed analysis of sensitivity to parameterization choice and vertical resolution using comparative 1D RANS simulations, and (4) a regional comparison between New England and Florida that contrasts meteorological regimes and their effect on forecast reliability.

This paper is structured as follows: Section 2 describes the NAM model configuration and the MYJ PBL scheme used. Section 3 details our methodology, including data processing and stability-based classification techniques. Section 4 analyzes

hub height in relation to key atmospheric boundaries across seasons and daily cycles in both regions. Section 5 examines forcing mechanisms and their impact on hub-height winds, focusing on differences between New England and Florida. Section 6 evaluates NAM parameterization and vertical resolution uncertainties specifically at hub height, incorporating comparative 1D RANS simulations to isolate these effects under controlled conditions. Conclusions will be drawn in Section 7.

## 2  Model Description

The North American Mesoscale (NAM) Forecast System (Colbert et al., 2019) is generally used for numerical weather prediction and encapsulates multiple domains over the North American continent at various horizontal resolutions. It solves the fully compressible and non-hydrostatic equations on the Arakawa E-grid (using the WRF-NMM model configuration) with a hybrid sigma-pressure vertical coordinate (Janjic, 2003). The model uses a forward-backward scheme for horizontally propagating fast waves, an implicit scheme for vertically propagating sound waves, and conserves multiple first and second-order

quantities including energy and enstrophy (Skamarock et al., 2008). The NAM model relies on a three-dimensional variational

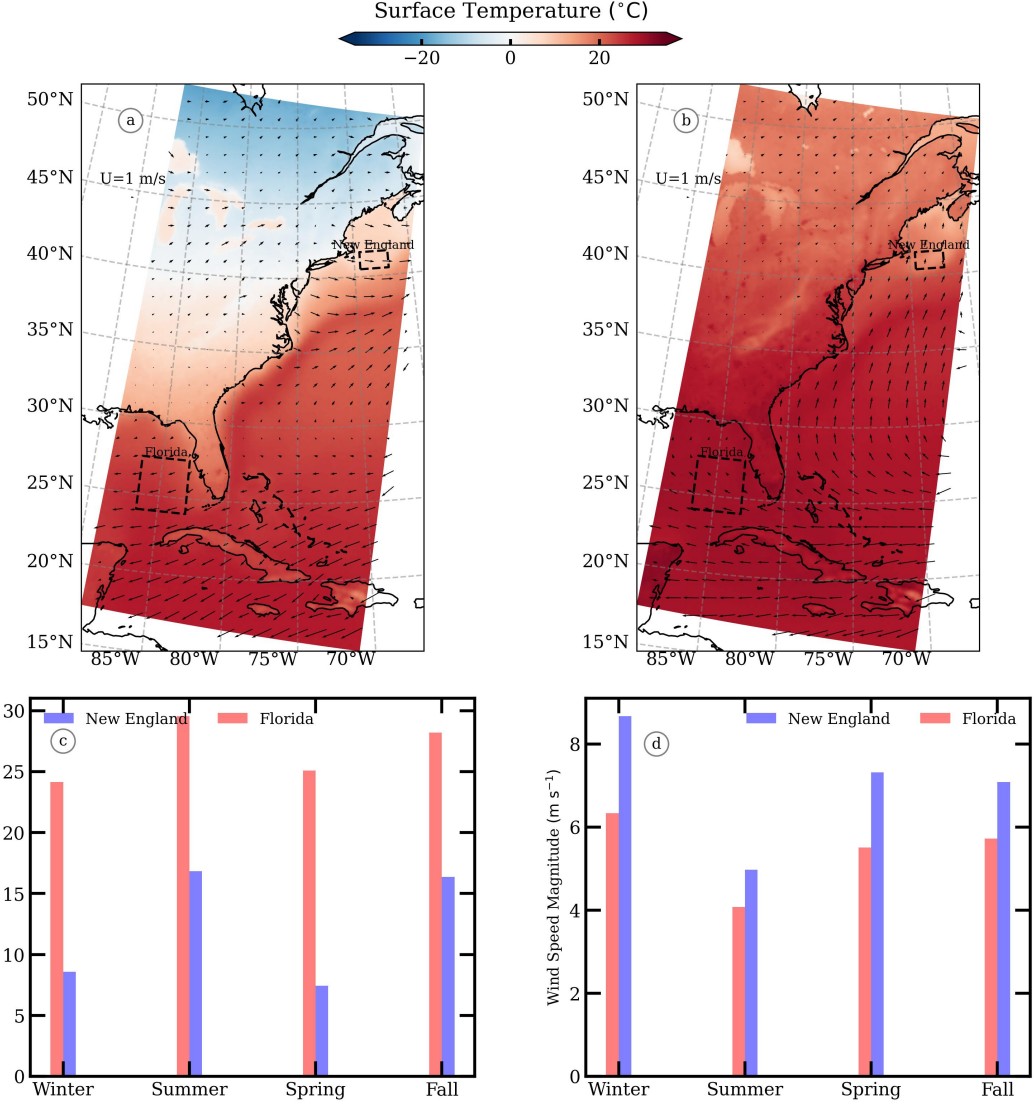

**Figure 1.** (a) Winter and (b) summer mean surface temperature distributions with overlaid 10-m wind vectors across the US east coast. Dashed boxes indicate the New England and Florida study regions. (c) and (d) seasonal surface temperatures and wind speed magnitude at the hub height (100 m) in both regions.

(3DVAR) assimilation system that incorporates observations from various sources, including radiosondes, aircraft, satellites, surface stations, and radar data (Rogers et al., 2009). The model has comprehensive physics options for land-surface processes, planetary boundary layer, radiation, microphysics, and cumulus convection, with a native horizontal resolution of 12 km and 45 vertical levels (Mesinger et al., 2012). The New England coastal area and the Florida coastal region were analyzed in this study (Figure 1). The NAM reanalysis data spans from 2012 through 2022 with a 6-hourly temporal resolution.





The model employs the Mellor-Yamada-Janjic (MYJ) turbulence closure scheme, a 1.5-order local closure model that parametrizes turbulent kinetic energy (TKE) to represent vertical mixing processes (Janjic, 2001). The MYJ scheme is based on the following TKE equation:

$$\frac{de}{dt} - \frac{\partial}{\partial z}(l_m\sqrt{2e}S_e\frac{\partial e}{\partial z}) = P_s + P_b - \epsilon \tag{1}$$

where the first term on the left-hand side represents the total derivative of turbulent kinetic energy (e) and the second term represents the vertical redistribution of e. The terms on the right side represent the shear production, buoyancy production, and dissipation, respectively. These terms can be written as follows:

$$P_s = -\overline{u'w'}\partial_z u - \overline{v'w'}\partial_z v \tag{2}$$

$$P_b = \beta\overline{w'\theta_v'} \tag{3}$$

$$\epsilon = \frac{(2e)^{3/2}}{B_1 l_m} \tag{4}$$

The Reynolds stresses can be related to a diffusive approximation of vertical mixing as follows:

$$-\overline{u'w'} = K_m\partial_z u \tag{5}$$

$$-\overline{v'w'} = K_m\partial_z v \tag{6}$$

$$-\overline{w'\theta_v'} = K_h\partial_z\theta_v \tag{7}$$

The vertical viscosity and diffusivity are scaled according to parameters as:

$$K_m = l_m\sqrt{2e}S_m \tag{8}$$

$$K_h = l_m\sqrt{2e}S_h \tag{9}$$





$S_m$, $S_h$, $\beta$ and $B_1$ are constants that are defined using experimental data and internal relations as described in Mellor and Yamada (1982) and Janjic (2001).

Within the PBL, the master mixing length $l_m$ can be formulated as follows:

$$l_m = l_0 \frac{\kappa z}{\kappa z + l_0} \tag{10}$$

where $l_0$ is a turbulent-velocity-weighted average height:

$$l_0 = \frac{\int_0^{z_i} z\sqrt{2e}\,dz}{\int_0^{z_i} \sqrt{2e}\,dz} \tag{11}$$

Close to the surface, $l_m \to \kappa z$ where $\kappa$ is the von Kármán constant (0.4). Between the top of the surface layer and the PBL height $z_i$, $l_m \sim l_0$. Above the PBL, the master mixing length $l_m \sim 0.23\delta z$ where $\delta z$ is the model vertical grid spacing. In this scheme, the PBL height is determined as the height where the turbulent kinetic energy falls below a critical threshold of 0.1 $m^2\ s^{-2}$ (Janjic, 2001). This MYJ-specific threshold represents the level where TKE production can no longer balance dissipation, identifying the transition from turbulent boundary layer to stratified free-atmosphere. This turbulence-based definition reflects a fundamental characteristic of the planetary boundary layer: the height at which turbulent mixing becomes insufficient to overcome stratification. In the atmosphere, this turbulent region is termed "mixing layer", where eddies redistribute momentum, heat, and moisture, as distinct from the "mixed layer" that results from this process (Holtslag and Nieuwstadt, 1986; Wyngaard, 2010; Fox-Kemper et al., 2022).

The MYJ parametrization is particularly important for our study as it directly influences the representation of vertical mixing processes in the atmospheric boundary layer (Janjic, 2003). The scheme's treatment of turbulent kinetic energy affects the simulation of wind profiles at hub height, which is crucial for characterizing wind resources in coastal environments. In these regions, atmospheric conditions are influenced by complex air-sea interactions and varying terrain features that can significantly impact wind speed and direction throughout the turbine rotor layer (Mesinger et al., 2012). The NAM forecast system uses the MYJ parameterization at relatively coarse vertical resolution: in Section 6 we compare this parameterization against others and evaluate its sensitivity to vertical resolution.

## 3 Methodology

### 3.1 Boundary Layer Stability Classification from Surface Fluxes

The atmospheric boundary layer can be classified as stable, neutral, or convective. These regimes can be categorized using the Obukhov length scale ($L_{MO}$), which characterizes the relative importance of shear and buoyancy in the surface layer (Monin and Obukhov, 2009). This parameter can be defined as:

$$L_{MO} = -\frac{u_*^3 \theta_v}{\kappa g \overline{w'\theta_v'}} \tag{12}$$



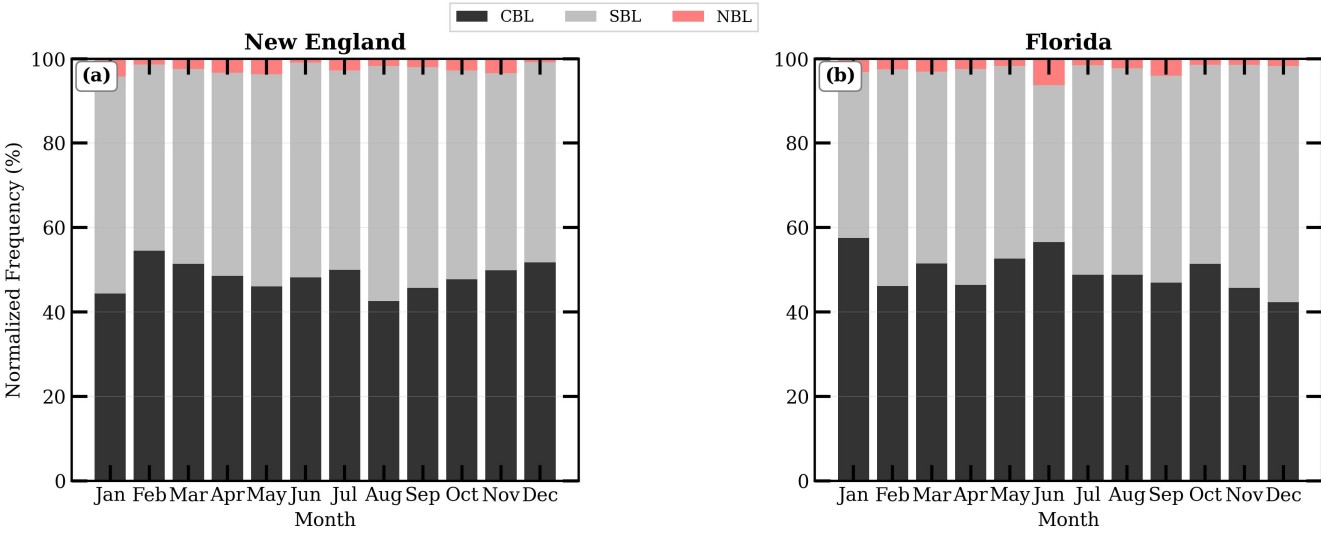

**Figure 2.** Monthly distribution of convective (CBL), stable (SBL), and neutral (NBL) boundary layer conditions for (a) New England and (b) Florida.

where $u_*$ is the friction velocity, defined as $u_* = \left(\frac{\tau}{\rho}\right)^{\frac{1}{2}}$ where $\tau$ is the wind stress and $\rho$ is the air density. This surface friction velocity characterizes the turbulent momentum at the surface and provides the velocity scale used throughout our analysis and at all altitudes, including the normalized wind profiles in Figure 3. $\theta_v$ is the virtual potential temperature (which approximates the combined buoyancy effects of humidity and temperature), $\kappa$ is the von Kármán constant (0.4), $g$ is the gravitational acceleration,

and $\overline{w'\theta_v'}$ is the kinematic virtual potential temperature flux combining the surface evaporation and heat flux. The direction of this temperature flux distinguishes between the convective and stable conditions, while the magnitude of this lengthscale weighs the relative importance of shear versus buoyancy.

Following previous studies (Gryning et al., 2007), we classified the boundary layer into three stability regimes:

– Convective Boundary Layer (CBL): $-500 < L_{MO} < -10$ m

– Neutral Boundary Layer (NBL): $|L| > 500$ m

– Stable Boundary Layer (SBL): $10 < L < 500$ m

These thresholds were applied to the NAM reanalysis data at each 6-hour interval for both the New England and Florida regions (Figure 1) throughout the study period. Surface flux parameters $(u_*, \overline{w'\theta_v'})$ essential for computing the Obukhov length scale were extracted directly from the NAM output fields.

Figure 2 shows the monthly distribution of these regimes for both regions. The analysis reveals different seasonal and local patterns in boundary layer stability. In both regions, the stable and convective boundary conditions predominate throughout the year, accounting for more than 90% of all cases in New England and Florida.





Regional differences exist in the seasonal distribution of stability regimes. New England exhibits a peak in stable conditions during August, with nearly 55% frequency, while in Florida the maximum number of stable cases is reached in December.

Convective conditions occur in both regions throughout the year, with fall/winter months (October through February) showing relatively high occurrence in both New England and Florida. Meanwhile, in summer, convection is stronger in Florida due to persistent surface heating in the subtropical climate. Neutral conditions maintain a relatively consistent presence throughout the years in both regions (less than 10% frequency).

These different regimes form the foundation for subsequent analyses of wind characteristics and power estimation as they

influence the vertical structure of wind profiles that interact with wind turbine rotors. Due to the limited occurrence of neutral boundary layer conditions in both regions and their less significant impact on wind power variability, the subsequent analysis will focus primarily on the stable and convective regimes.

### 3.2 Thermodynamic Analysis of Atmospheric Stratification

Atmospheric thermodynamic conditions can be evaluated using Convective Available Potential Energy (CAPE) and Convective

Inhibition (CIN) parameters (Riemann-Campe et al., 2009). These parameters sketch the potential for background conditions to form convective turbulence, which influences boundary layer characteristics.

CAPE represents the amount of buoyant energy available to accelerate air parcels upward and is calculated as:

$$\text{CAPE} = \int\limits_{LFC}^{EL} g \frac{T_{parcel} - T_{env}}{T_{env}} dz \tag{13}$$

"where $LFC$ is the level of free convection, $EL$ is the equilibrium level, $g$ is gravitational acceleration, $T_{parcel}$ is the

temperature of a lifted air parcel, and $T_{env}$ is the environmental temperature. The Level of Free Convection (LFC) represents the height where atmospheric parcels become buoyant and rise freely, initiating convective turbulence that enhances vertical mixing across wind turbine rotors. The Equilibrium Level (EL) marks the upper boundary where this convective mixing ceases. Between these levels, enhanced turbulent mixing is expected to affect wind turbine wake recovery and power fluctuations directly. While the dynamics of the LFC-EL layer are distinct, the effects of strong convective mixing on turbines have been

quantified in other atmospheric layers. For instance, in a surface-driven daytime convective boundary layer, stronger vertical momentum transport was found to increase power output variability by 15-30% and accelerate wake dissipation (Wharton and Lundquist, 2012).

Conversely, CIN quantifies the energy barrier that must be overcome before convection can occur:

$$\text{CIN} = \int\limits_{SFC}^{LFC} g \frac{T_{parcel} - T_{env}}{T_{env}} dz \tag{14}$$

where $SFC$ represents the surface level.



CAPE and CIN values were extracted directly from the NAM reanalysis dataset at 6-hour intervals for both the New England and Florida coastal regions throughout the study period. Joint probability density functions (PDFs) were constructed to characterize the relationship between these parameters. This approach allows for statistical representation of co-occurring CAPE and CIN values and their frequency of occurrence within the dataset. Two types of joint PDFs were evaluated:

– Seasonal joint PDFs of CAPE and CIN: Data were stratified by summer (JJA) and winter (DJF) seasons.

   – Seasonal joint PDFs of diurnal ratios: To quantify diurnal variations, we calculated the ratios $\frac{CAPE_{night}}{CAPE_{daytime}}$ and $\frac{CIN_{night}}{CIN_{daytime}}$, where nighttime values were taken from 00 UTC (midnight) analyses, and daytime values from 06 UTC (6 AM) analyses.

### 3.3   Power Estimation and Sensitivity Analysis

The power production uncertainty under different atmospheric stability was evaluated using a one-dimensional Reynolds-
Averaged Navier-Stokes (RANS) model for the planetary boundary layer (Kanoksirirath, 2023). 1D RANS simulations are used to isolate the effects of different parameterization schemes and vertical resolution on hub-height winds. This approach allows controlled sensitivity experiments that would be computationally prohibitive with 3D simulations. The 1D framework is appropriate here because we focus on vertical profiles and boundary layer structure rather than horizontal wake effects. The model solves the governing equations for conservation of momentum and heat:

$$\frac{\partial U}{\partial t} = f(V - V_g) - \frac{\partial \overline{u'w'}}{\partial z} \tag{15}$$

$$\frac{\partial V}{\partial t} = -f(U - U_g) - \frac{\partial \overline{v'w'}}{\partial z} \tag{16}$$

$$\frac{\partial \Theta}{\partial t} = \frac{d\Theta}{dt}\Big|_{rad} - \frac{\partial \overline{w'\theta'}}{\partial z} \tag{17}$$

where $U$ and $V$ are the mean wind components, $U_g$ and $V_g$ are the geostrophic wind components, $\Theta$ is the potential temperature, $f$ is the Coriolis parameter, and $\overline{u'w'}$, $\overline{v'w'}$, and $\overline{w'\theta'}$ are the turbulent momentum and heat fluxes, respectively.

The turbulent fluxes were parameterized using different closure schemes, with the relationship:

$$\overline{u'w'} = -K_m \frac{\partial U}{\partial z}, \quad \overline{v'w'} = -K_m \frac{\partial V}{\partial z}, \quad \overline{w'\theta'} = -K_h \frac{\partial \Theta}{\partial z} \tag{18}$$

where $K_m$ and $K_h$ are the eddy diffusivity coefficients for momentum and heat, respectively.

The Monin-Obukhov similarity theory was applied for both momentum and temperature profiles:

$$U(z) = \frac{u_*}{\kappa}\left[\ln\left(\frac{z}{z_0}\right) - \psi_m\left(\frac{z}{L_{MO}}\right)\right] \tag{19}$$





$$\Theta(z) - \Theta_0 = \frac{\theta_*}{\kappa} \left[ \ln\left(\frac{z}{z_0 h}\right) - \psi_h\left(\frac{z}{L_{MO}}\right) \right] \tag{20}$$

where $\kappa$ is the von Kármán constant (0.4), $z_0$ and $z_{0h}$ are the roughness lengths for momentum and heat respectively, $\Theta_0$ is the surface potential temperature, and $\psi_m$ and $\psi_h$ are the stability functions for momentum and heat.

For stable conditions ($L_{MO} > 0$):

$$\psi_m\left(\frac{z}{L_{MO}}\right) = -4.7\frac{z}{L_{MO}} \tag{21}$$

$$\psi_h\left(\frac{z}{L_{MO}}\right) = -4.7\frac{z}{L_{MO}} \tag{22}$$

For unstable conditions ($L_{MO} < 0$):

$$\psi_m\left(\frac{z}{L_{MO}}\right) = 2\ln\left(\frac{1+x}{2}\right) + \ln\left(\frac{1+x^2}{2}\right) - 2\tan^{-1}(x) + \frac{\pi}{2} \tag{23}$$

$$\psi_h\left(\frac{z}{L_{MO}}\right) = 2\ln\left(\frac{1+x^2}{2}\right) \tag{24}$$

where $x = (1 - 15z/L_{MO})^{1/4}$.

A reference simulation was set with the following parameters: time step $\Delta t = 0.01$ s, friction velocity $u_* = 0.3$ m/s, a vertical grid spacing $\Delta z = 10$ m, and a zonal wind with $u(t=0) = u_g = 8m\ s^{-1}$, and the Mellor-Yamada-Janjić (MYJ) turbulence closure scheme (Janjic, 2001). This closure scheme is used to parameterize the vertical viscosity and diffusivity defined in equations 8 and 9. For stable conditions, the surface temperature scale was set to $\theta_* = 0.01$ K, Obukhov length $L_{MO} = 100$ m, and cooling rate $d\Theta/dt = 0.5$ K/h. For unstable conditions, $\theta_* = -0.01$ K, $L_{MO} = -100$ m, and heating rate $d\Theta/dt = -0.5$ K/h. All the simulations were run for 24h with hourly outputs.

Based on this reference configuration, we conducted a series of sensitivity experiments as summarized in Table 1. Note that while these conditions are not directly drawn from the NAM conditions, the rate of heating, $\theta_*$, and $L_{MO}$ are within the range of conditions visited by the NAM. In this way, running a 1D model with the different parameter and parameterization choices in Table 1 provide an estimate of the power uncertainty in the conditions found in the NAM.

At each time step, the wind power density at hub height (100 m) was calculated using:

$$P = \frac{1}{2}\rho U^3 C_p \tag{25}$$

where $\rho$ is the air density, $U$ is the horizontal wind speed at hub height, and $C_p$ is the Betz coefficient (power coefficient) with a theoretical maximum value of 0.593. This coefficient represents the maximum fraction of the wind's kinetic energy that





**Table 1.** Summary of the reference simulation (stable and unstable) and sensitivity simulations

| Simulation | Closure | $\Delta z$ (m) | $\theta_*$ (K) | $L_{MO}$ (m) | $d\Theta/dt$ (K/h) |
|---|---|---|---|---|---|
| Reference | MYJ (Janjic, 2001) | 10 | ± 0.01 | ± 100 | ± 0.5 |
| KPP (stable) | KPP (Large et al., 1994) | 10 | 0.01 | 100 | 0.5 |
| KPP (unstable) | KPP | 10 | -0.01 | -100 | -0.5 |
| K-epsilon (stable) | K-epsilon (Launder and Spalding, 1974) | 10 | 0.01 | 100 | 0.5 |
| K-epsilon (unstable) | K-epsilon | 10 | -0.01 | -100 | -0.5 |
| Coarse grid (stable) | MYJ (Janjic, 2001) | 100 | 0.01 | 100 | 0.5 |
| Coarse grid (unstable) | MYJ | 100 | -0.01 | -100 | -0.5 |
| Neutral | MYJ | 10 | 0 | 1000 | 0 |

can be extracted by an ideal wind turbine, as established by Betz's law. For our analysis, we adopted the Betz limit as the

power coefficient to represent the theoretical maximum extractable power. The Betz limit (Cp = 0.593) is adopted here rather than specific turbine curves because our focus is on relative power density differences between parameterization schemes, not absolute power production. This approach isolates atmospheric effects from turbine-specific characteristics. Using a standard turbine model would introduce additional assumptions about cut-in/cut-out speeds and rated power that are not central to our analysis of boundary layer parameterization uncertainties.

To quantify the uncertainty in power estimation related to model configuration, we calculated the relative power density difference compared to the reference simulation ($\Delta P_{\text{rel}} = \frac{P - P_{\text{ref}}}{P_{\text{ref}}} \times 100\% \approx 3\frac{U - U_{\text{ref}}}{U_{\text{ref}}} \times 100\%$) and the temperature deviation at hub height ($\Delta T = \frac{T - T_{\text{ref}}}{T_{\text{ref}}} \times 100\%$). This methodology allows for a systematic assessment of uncertainties in wind power estimation related to the representation of the planetary boundary layer in numerical models under varying atmospheric stability conditions.

## 245   4   Hub Height Relation to Boundary Layer Structure: Regional and Seasonal Patterns

### 4.1   Vertical Structure of the Atmospheric Boundary Layer

The atmospheric boundary layer (ABL) under stable and convective regimes during winter (DJF) and summer (JJA) is sketched in Figure 3. In New England and Florida, the temperature distribution ($\frac{\theta}{\theta_{hub}}$) in the convective regime is nearly uniform with height, highlighting the well-mixed nature of unstable conditions. In New England (Figure 3a), subtle seasonal differences

in the convective regime, marked by greater stratification, can be observed in winter. The stable boundary layer displays stronger stratification (increasing $\frac{\theta}{\theta_{hub}}$ with height), characterizing limited/suppressed vertical mixing. Meanwhile, in Florida (Figure 3c), no noticeable difference between the temperature structure can be observed throughout the season, suggesting more uniform thermal forcing. These temperature profile differences can have potential implications for turbine performance metrics. As Bodini et al. (2021) demonstrated, atmospheric stability conditions can substantially influence wind turbine wakes and

power output. In stable conditions, the stronger stratification observed in New England would likely result in more persistent





wakes that could extend further downstream, potentially affecting power production in wind farms. According to Kim et al. (2021), these stratification differences can cause variations in power curves of up to 200 kW at the same wind speed, with the strongest effects observed in stable conditions.

In winter, CBL conditions in New England produce the strongest wind shear ($\frac{U}{u^*}$) reaching approximately 60 at 1.2 km height, while summer SBL conditions show the weakest shear (Figure 3b). This substantial seasonal variation in New England contrasts markedly with Florida (Figure 3d), where the wind shear remains relatively consistent across seasons with considerably lower magnitudes ($\sim 20$). This reflects major differences in synoptic forcing and thermal gradients between the regions. These regional wind profile differences significantly affect wind power density calculations and turbine selection criteria. Marini et al. (2025) shows that differences in wind shear across the rotor plane impact both the annual energy production and

the mechanical loading on turbine components. The higher wind shear in New England during winter CBL conditions would likely necessitate turbines designed to handle greater mechanical loads across the rotor disk, while also potentially yielding higher energy production due to stronger winds at hub height. Kara and Şahin (2023) noted that these regional variations in wind profiles require location-specific approaches to turbine design and selection to optimize performance while ensuring structural integrity.

These conditions may affect the statistical distribution of PBL heights across both regions (Figure 3e and f). In New England, the PBL heights distribution spans a broader range with significant overlap between winter and summer, regardless of the stability regime (Figure 3e). The CBL heights extend to approximately 2 km in summer, while SBL heights show similar ranges across seasons. In Florida (Figure 3f), a clearer separation between summer and winter PBL heights can be observed, particularly in the SBL, where winter stable layers are globally shallower than summer ones. These statistical distributions

have important implications for uncertainty in wind resource assessments. The boundary layer height variations directly affect the vertical extent of wind turbines' wakes, with implications for energy predictions. The broader distributions in New England suggest higher uncertainty in resource assessment and a need for high-fidelity modeling approaches in this region. As mentioned by Marini et al. (2025), these variations necessitate careful consideration when employing standard power-law profiles for wind resource assessment, as significant deviations can occur, particularly during stable conditions or when turbines operate

near the boundary layer top.

### 4.2    Seasonal Variation in Hub Height to Dynamical Heights Ratio

The seasonal variation in hub height to PBL height ratios in New England and Florida under CBL and SBL conditions reveals interesting regional differences (Figure 4a and b). In New England, the ratio $\frac{H_{hub}}{H_{pbl}}$ depicts a monthly variability with a peak in July, where the mean exceeds 2 under CBL conditions, indicating that turbine hubs frequently operate above the shallow atmo-

spheric boundary layer. This anomaly indicates that turbines may operate above the PBL, leading them to encounter different flow regimes characterized by reduced turbulence and different wind shear patterns. For example, the Monin-Obukhov similarity functions are *not* expected to apply outside of the PBL, even when $|L_{MO}|$ is large and shear dominates. When turbines operate near or above the boundary layer top, they can experience different atmospheric conditions that affect their downstream wake characteristics Bodini et al. (2021). Indeed, the vertical wind speed deficit may be stronger and more persistent in such

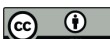

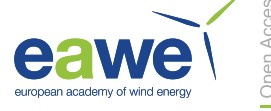 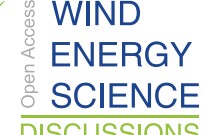

**Figure 3.** NAM normalized potential temperature profiles (a,c) and non-dimensional wind profiles (b,d) with their respective error bars for New England and Florida under different stability conditions and seasons. Bottom panels (e,f) show violin plots of planetary boundary layer heights for both stability regimes across winter and summer seasons.





conditions. In contrast, the turbines operate consistently within the boundary layer regardless of the season in Florida, as their hub-to-PBL height ratios remain below 0.5 throughout the year (Figure 4b).

The Obukhov length scale ($|L_{MO}|$) may also impact the turbines at the hub under different regimes (Figure 4c and d). Higher ratios under SBL conditions are observed during summer in New England (values frequently exceeding 4) compared to Florida. This indicates that turbines may operate under a stronger stability effect in the New England region. The atmospheric stability regimes significantly influence annual energy production (AEP), with variations of 1.4-4% depending on the stability conditions (Kim et al., 2021). The higher $\frac{H_{hub}}{|L_{MO}|}$ in New England during summer suggests that turbines in this region will experience buoyancy forcing stabilizing effects that will influence their summer and thus annual energy production.

These regional variations align with findings from Abraham et al. (2024), who observed significant increases in PBL height downstream of wind plants under stable conditions, but only when the upstream PBL height was shallow (less than 0.25 km). The thermodynamic measurements from the AWAKEN campaign showed PBL height increases of 33-39% in these conditions, while turbulence-based measurements indicated even larger increases of up to 141%. This relationship between ambient PBL height and wind plant effects helps explain why regions with seasonally variable PBL height, like New England, might experience more pronounced wind plant-atmosphere interactions during certain times of the year. Lavers et al. (2019) further corroborate this relationship, showing that forecast errors in PBL height over oceans are larger in the midlatitudes and under unstable PBL conditions, highlighting regional variability in boundary layer predictability. The complexity of these interactions is particularly relevant for New England's offshore wind development, including operational projects like the South Fork Wind Farm. Recent large eddy simulations of this facility have revealed how nocturnal boundary layer dynamics significantly influence wake dynamics and turbulence production (Ayouche et al., 2025), highlighting the importance of accurately characterizing regional atmospheric conditions as presented in this study.

Additionally, Bodini et al. (2019) found that offshore turbulence dissipation rates are typically two orders of magnitude smaller than onshore, with a much more subtle diurnal cycle, indicating weaker dissipation, transfer and production of turbulence, for generally weaker turbulence for a given wind speed. This reduced turbulence means that wind plant wakes will be stronger and persist farther downwind in summer offshore environments. This finding is particularly relevant for the regional differences observed between New England and Florida, as the different maritime influences and coastal geometries may contribute to the distinct seasonal patterns in PBL height and stability metrics observed in these regions.

### 4.3 Diurnal Evolution of the Planetary Boundary Layer

The diurnal evolution of the PBL height in New England shows significant day-night differences across the annual cycle (Figure 5). For each day in our dataset, we calculate the difference between the 00 UTC (nighttime) and 12 UTC (daytime) PBL heights. These differences are then averaged by month and stability regime. This evolution reveals an important nocturnal PBL collapse throughout the year, with the most notable differences occurring during spring and summer months (March through June). During these months, the nighttime PBL can be 400-450 meters shallower than the daytime boundary layer under both stability regimes. The physical mechanisms driving these seasonal differences in diurnal PBL evolution relate to the varying intensity of solar radiation and surface heating throughout the year. As explored by Marini et al. (2025), these





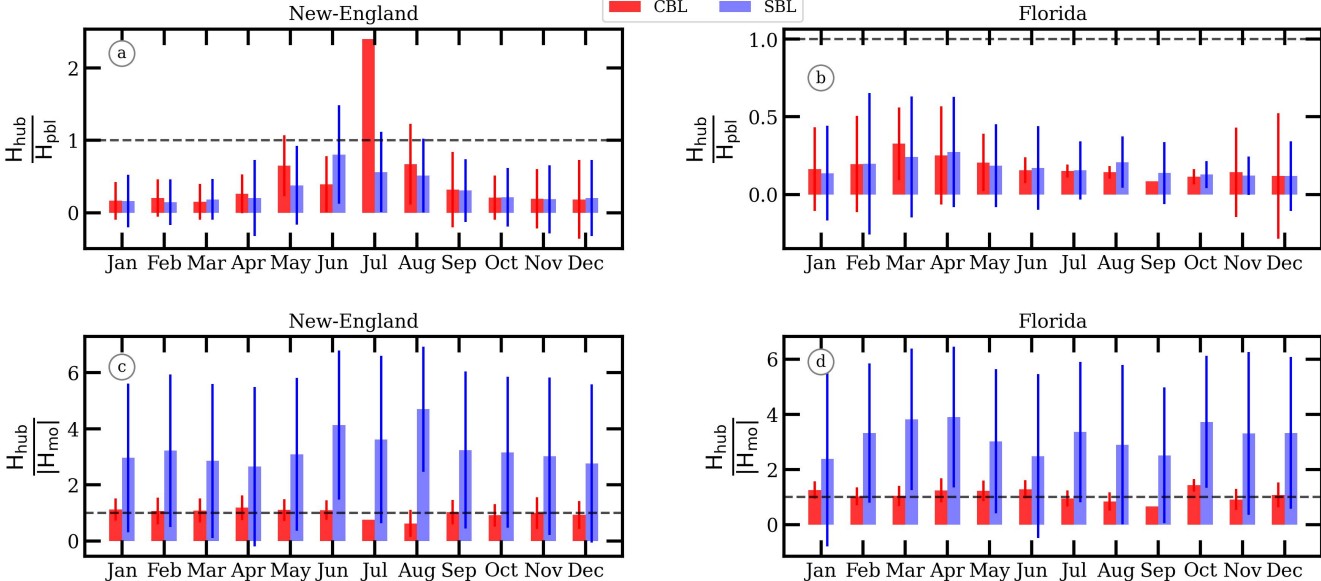

**Figure 4.** Monthly distributions of the ratio of hub height to planetary boundary layer height (a,b) and hub height to Obukhov length (c,d) for New England and Florida, categorized by convective (CBL, red) and stable (SBL, blue) boundary layer conditions.

diurnal transitions create complex wind profiles that deviate significantly from the standard power law assumptions often used in wind resource assessment. In winter (November through February), moderate diurnal variations are observed with night differences typically less than 250 m. The SBL generally exhibits similar or slightly deeper nocturnal variations compared to the convective regime, particularly during the transitional seasons.

This diurnal restructuring of the boundary layer contributes to significant challenges in wind resource forecasting, as the relative position of hub height to the evolving PBL structure directly impacts wind shear, turbulence intensity, and ultimately power production efficiency throughout the daily cycle. Potential improvements to numerical weather prediction models to better capture these diurnal transitions are essential. As demonstrated by Kim et al. (2021), incorporating accurate representations of atmospheric stability in power prediction models can significantly reduce forecast errors, especially during stable conditions when the boundary layer structure undergoes its most important changes. Recent observations from Abraham et al. (2024) revealed that wind plants can delay the evening transition from convective to stable conditions through enhanced mixing that counteracts the suppression of turbulence by the cooling surface. The strongest relative PBL height increases were observed around 01 UTC (early evening) and 07-08 UTC (middle of the night), precisely during critical transition periods and when the upstream boundary layer was at its shallowest.

The regional differences in boundary layer structure and their relationship to turbine hub heights demonstrate the need for accurate location-specific approaches to assess the wind power resources. As suggested by Marini et al. (2025), incorporating realistic wind profiles based on field measurements rather than relying solely on idealized power law profiles would





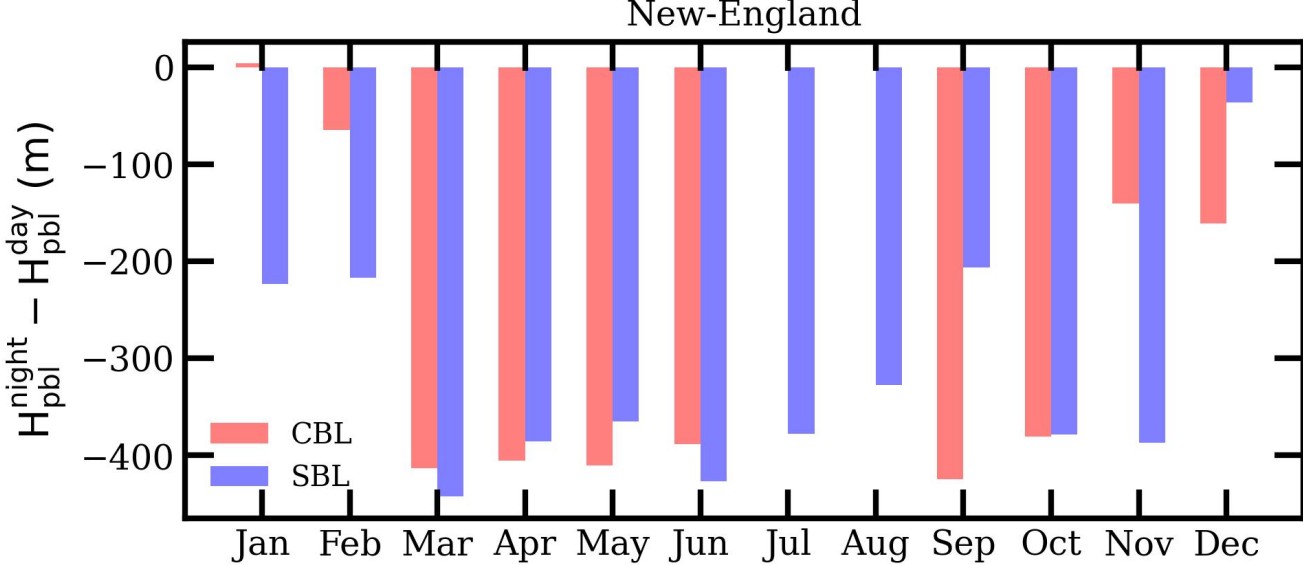

**Figure 5.** Monthly distribution of the average difference between nighttime and daytime planetary boundary layer heights in New England, separated by convective (CBL, red) and stable (SBL, blue) boundary layer conditions.

significantly improve the accuracy of both energy yield predictions and load simulations for wind turbines. This approach is especially important in light of findings Bodini et al. (2019) that offshore environments exhibit fundamentally different turbulence characteristics, which must be properly accounted for in modeling.

## 5 Forcing Mechanisms and Their Impact on Hub-Height Winds

### 5.1 Seasonal Variations in CAPE and CIN Regimes

The regional and seasonal differences in CAPE and CIN distributions between New England and Florida are sketched in Figure 6. In New England (Figure 6a), limited CAPE (rarely exceeding 500 $J\ kg^{-1}$) and moderate CIN values are observed during winter. This environment reflects predominantly stable atmospheric conditions that characterize the northeastern United States during winter months. However, in summer (Figure 6c), large CAPE values (exceeding 1500 $J\ kg^{-1}$) and stronger CIN are observed, suggesting favorable conditions for convection. Florida exhibits fundamentally different patterns (although wintertime in Florida somewhat resembles summertime in New England from this perspective). In winter (Figure 6b), moderate CAPE values are maintained, frequently exceeding 1000 $J\ kg^{-1}$ alongside modest CIN. The summer distribution (Figure 6d) shows an expansion toward higher CAPE values but with a more diffuse relationship to CIN than observed in New England.

These regional differences can also be illustrated in the independent distributions of CAPE and CIN (Figure 6e and f). The seasonal contrasts in CAPE under both convective boundary layer and stable boundary layer conditions can be observed in




New England, while in Florida, a consistently higher CAPE is maintained throughout the year. The consistently higher CAPE regime in Florida can lead to stormy, convective weather, which would result in gusty winds and turbulent air. These conditions suggest that Florida may not be an optimal location for wind farm development, as turbines would likely experience consistent reductions in power output and be subjected to significant mechanical stress on blades and structural components. The frequent
convective activity would increase maintenance requirements and potentially shorten turbine lifespans compared to regions with more moderate and predictable atmospheric conditions. Under climate change, New England is trending toward more convective storms (Kodero and Lee, 2025).

A study (Chen et al., 2020) found that CAPE increases almost everywhere under warming conditions, with the largest increases occurring in the ITCZ, western tropical Pacific, and summer continents. Their analysis showed that this seasonal
CAPE increase results primarily from increased low-level specific humidity, which leads to more latent heating and buoyancy for lifted parcels. These findings help explain the dramatic seasonal shift observed in New England's CAPE distribution, where summer humidity increases drive the substantial growth in CAPE values.

Seeley et al. Seeley and Romps (2015) explain through their "zero-buoyancy" theory that undiluted parcel buoyancy in the upper troposphere increases dramatically with warming temperatures, while remaining relatively insensitive in the lower
troposphere. This mechanism helps explain why the seasonal variations in CAPE are more pronounced in regions like New England that experience greater seasonal temperature swings compared to the more consistent tropical conditions of Florida.

## 5.2 Diurnal Evolution of CAPE and CIN Ratios

Diurnal variations in CAPE and CIN between New England and Florida are shown in Figure 7. In winter, New England (Figure 7a) exhibits notably high nighttime-to-daytime CIN ratios, frequently exceeding 20, indicating a strong suppression
of convection during nighttime hours. CAPE values during this period remain comparatively low, with ratios generally below 20, suggesting a predominantly stable nocturnal environment. Florida's winter pattern (Figure 7b) displays a broader spread in both CAPE and CIN ratios, though it similarly shows an enhancement of CIN at night.

More pronounced regional contrasts emerge during summer. In New England (Figure 7c), nighttime CIN is often 20 to 40 times stronger than daytime values, while nighttime CAPE is substantially reduced, typically reaching only 20–60% of
daytime levels. This pattern reflects a strong diurnal cycle with significant nocturnal stability. In contrast, Florida's summer pattern (Figure 7d) is characterized by a tighter distribution, with nighttime-to-daytime CAPE ratios rarely exceeding 40. This reduced variability is consistent with Florida's persistent marine tropical air mass, which limits the amplitude of diurnal thermodynamic changes.

These diurnal contrasts are further illustrated by the independent distributions in Figures 7e and 7f. Both regions show higher
daytime CAPE during summer months, while nighttime values reveal more complex behavior and overlap across seasons. Such patterns suggest that diurnal thermodynamic cycles play a significant role in shaping wind conditions that directly impact turbine operation and forecasting.



**Figure 6.** (a-d) The joint probability density functions (PDF) of Convective Available Potential Energy (CAPE) and Convective Inhibition (CIN) for winter and summer seasons in New England and Florida. (e,f) the seasonal and regional distributions of CAPE and CIN by stability regime.





Tuckman et al. (2023) identified a mechanism wherein strong nighttime boundary layer cooling generates substantial CIN, which suppresses early convection and facilitates moisture accumulation near the surface. This nocturnal inhibition process
likely contributes to the enhanced CIN observed in Figure 7, particularly in New England during summer.

For wind energy applications, these diurnal shifts have operational implications (Marini et al., 2025). Daytime conditions, with elevated CAPE and reduced CIN, are typically associated with increased turbulence, intermittent storms, and vertically mixed wind profiles. Nighttime periods, dominated by stronger CIN, often correspond to more stratified flow and enhanced wind shear, especially when low-level jets are present. These dynamics are more variable in New England due to its stronger

diurnal contrasts, whereas Florida's smoother transitions suggest a more predictable wind environment.

## 6   NAM Parameterization and Vertical Resolution Uncertainties at Hub Height

### 6.1   Comparison of Parameterized and Resolved Turbulence

Figure 8 shows the NAM parameterized TKE against resolved TKE (calculated as $0.5(u*^3 + w*^3)^{2/3}$) for New England under a convective regime–roughly, one would expect that resolved and parameterized TKE would co-vary. This analysis reveals a

moderate correlation between parameterized and resolved turbulence with $R^2$ values of 0.52 for summer and 0.45 for winter conditions. This low correlation points out systematic biases in the NAM model's representation of turbulence. At lower TKE values (below approximately $100 \ m^2 \ s^{-2}$, representing the 85th percentile of observed conditions in the New England marine atmospheric boundary layer during both winter and summer seasons), NAM tends to overestimate parameterized turbulence, while at higher values it underestimates turbulence intensity. While the 6-hourly temporal resolution of the NAM reanalysis data

inherently means that individual, rapid extreme events are not explicitly resolved, our analysis in this section demonstrates that this resolution is nevertheless sufficient to reveal fundamental and systematic biases within the Mellor-Yamada-Janjic (MYJ) turbulence parameterization. These biases, rooted in the scheme's physical formulation rather than solely data granularity, critically limit the model's capacity to accurately represent crucial turbulent structures and thus its ability to simulate extreme wind conditions, even if operating at finer temporal scales. This behavior creates challenges for wind energy applications,

as extreme turbulence events that significantly impact turbine loading and wake recovery may not be accurately captured. The analysis also highlights a greater spread in winter compared to summer, particularly at higher TKE values. This seasonal difference might be related to winter temperature contrasts and strong dynamical height variations.

In Reynolds-Averaged Navier-Stokes (RANS) or Large Eddy Simulation (LES) models, theory suggests that parameterized (subgrid) TKE should correlate strongly with the resolved TKE, assuming scale separation in the RANS case and equilibrium

turbulence conditions (SMAGORINSKY, 1963; Deardorff, 1980). In the LES case, the largest turbulence is expected to be resolved and so a functional relationship is expected which follows from the Kolmogorov hypothesis, where TKE cascades from large to small scales at a rate proportional to $\frac{TKE^{3/2}}{L}$, where L is the turbulent length scale Pope (2000). However, several factors complicate this relationship in the NAM model, which is in the RANS regime. In this regime, the resolved kinetic energy can act as a source for turbulence, e.g., by shear production. First, the resolved and parameterized TKE calculations

include convective forcing through $w_*$, derived from the Obukhov length, which can be overestimated or underestimated due to



**Figure 7.** (a-d) the joint probability density functions (PDF) of nighttime-to-daytime ratios for CAPE and CIN across different seasons and regions. (e,f) The diurnal distributions of CAPE and CIN by season and stability regime.





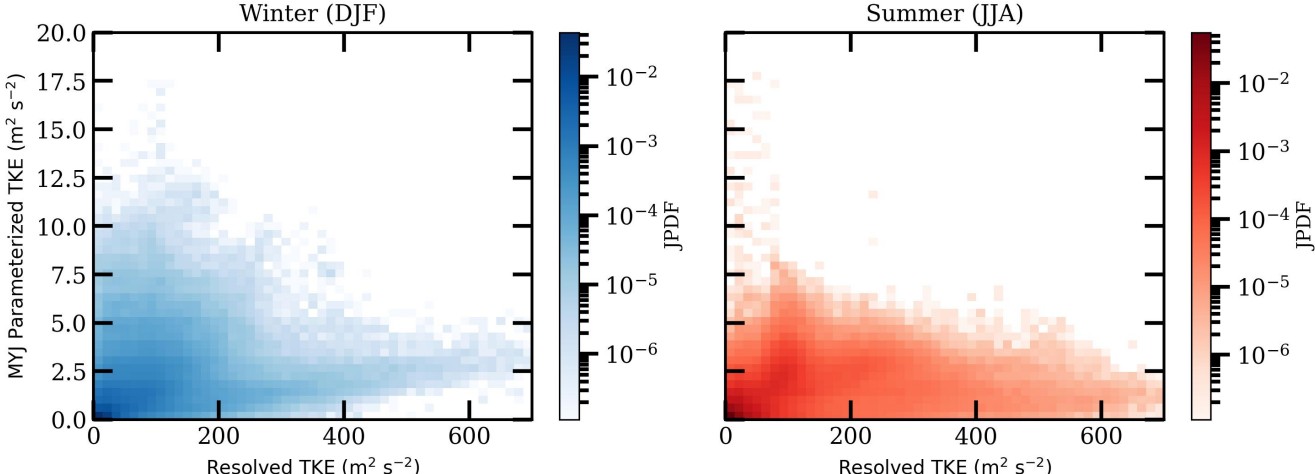

**Figure 8.** Joint PDF showing the relationship between resolved turbulent kinetic energy (TKE) and the Mellor-Yamada-Janjic (MYJ) parameterized TKE for New England during winter (left) and summer (right) seasons.

uncertainties in surface flux parameterizations (LeMone et al., 2008) and may generate parameterized TKE (i.e., parameterized convection) without generating significant resolved convection. Second, the coarse vertical resolution in operational models like NAM (with typical vertical spacing of 50-100 m in the boundary layer) inadequately resolves the sharp gradients near the surface and at the PBL top, leading to systematic biases in both resolved and parameterized components (Noh et al., 2003).

Finally, the MYJ scheme's local closure assumption may not capture non-local transport processes that become important in convective conditions, further degrading the correlation (Wyngaard, 2004). The discrepancy of resolved and parameterized TKE suggests that care is needed in the boundary layer parametrization scheme and vertical resolution in mesoscale models for them to be suitable for wind energy forecasting.

Wu et al. (2022) found similar challenges in their sensitivity study of WRF parameterizations, noting that the performance

of PBL schemes varies significantly with both terrain complexity and atmospheric stability conditions. They observed that the Yonsei University (YSU) scheme generally provided the most consistent predictions across varied conditions, but the performance of all schemes was dependent on the resolution of the underlying topography dataset.

These two basic evaluations of parameterization skill stimulated further investigation into sensitivity of hub-height statistics on parameterization chosen from some similar to those in NAM and vertical resolution. The next sections detail how the

uncertainty in forecasts of hub-height conditions stems from these model challenges.

## 6.2 Impact of Model Configuration on Hub-Height Winds

To isolate the effects of different parameterization schemes and vertical resolution under controlled conditions, we conducted a series of one-dimensional RANS simulations of the planetary boundary layer (Figures 9, 10 and 11). This method was inspired by the design of our similar study on oceanic boundary layer sensitivity to mixing schemes (Li et al., 2019).





The time series of hub-height wind speed (Figure 9a) reveals significant differences between stable, unstable, and neutral atmospheric conditions. Under stable stratification, the simulations capture the development of a pronounced low-level jet feature, with wind speeds approaching 10 m/s at certain periods. In the stable case, particularly evident in Figure 9a during hours 12-14, a low-level jet (LLJ) is indicated by the maximum peak in hub-height wind speeds around 10 m/s. An LLJ is typically defined as a local wind speed maximum within the lowest few hundred meters of the atmosphere (generally below 500 m), characterized by a notable decrease in wind speed both above and below its core (Weide Luiz and Fiedler, 2022; Hallgren et al., 2023). The LLJ can also be observed in Figure 10a more explicitly, demonstrating the required wind speed falloff above and below the jet core, which signifies a flow decoupled from surface friction within a stably stratified boundary layer. This jet structure, frequently observed in offshore environments under stable conditions, creates significant vertical wind shear across the rotor layer that cannot be adequately represented by coarse resolution models. The analysis period hereafter is within the time window of 8 to 18 hours.

The normalized velocity profiles (Figure 10a) further illustrate how different turbulence closure schemes produce varying vertical wind structures. The stable case exhibits significantly stronger shear across the rotor layer, with a pronounced inflection point around 200m height. For modern large-diameter turbines with blades spanning more than 200 meters, this means that different portions of the rotor disk experience substantially different wind conditions simultaneously, creating complex loading patterns not captured in simpler models. The temperature profiles (Figure 10b) demonstrate the corresponding thermal stratification patterns that drive these wind profile differences. Under stable conditions, the temperature gradient acts to suppress turbulent mixing, allowing stronger shears to develop and maintain. Under unstable conditions, enhanced turbulent mixing creates more uniform vertical profiles. The vertical structure of the ABL is also sensitive to closure schemes and vertical resolution, with major differences across the stable regime. In this regime, the wind speed/temperature is overall overestimated/underestimated, reaching a higher uncertainty in KPP closure scheme and coarser resolution.

Most significantly, the power density deviation results (Figure 11a) quantify how these model configuration choices translate into uncertainty in wind power estimates. The K-epsilon turbulence scheme produces power estimates approximately 7% higher than the reference MYJ scheme under unstable conditions, while predicting 10-15% lower power under stable conditions. These differences highlight how turbulence closure scheme selection alone can introduce significant uncertainty in power production estimates.

Vertical resolution also emerges as a critical factor, with coarser grid spacing leading to power density deviations of approximately 10-20% depending on stability conditions. This finding is particularly relevant for operational forecast models like NAM, which typically employ relatively coarse vertical resolution compared to large eddy simulations and typical hub heights. The neutral reference case shows the smallest deviations with the unstable case compared to the stable regime, where the uncertainty reaches 20%. The predominance of neutral-condition studies in wind tunnel experiments and simplified wake models can lead to strong biases regarding turbine performance, especially expectations of performance under stable conditions.

These findings align with previous studies (Marini et al., 2025), who found that model configuration choices significantly impact the representation of coastal and offshore wind profiles. Similarly, Siuta et al. (2017) demonstrated that hub-height wind




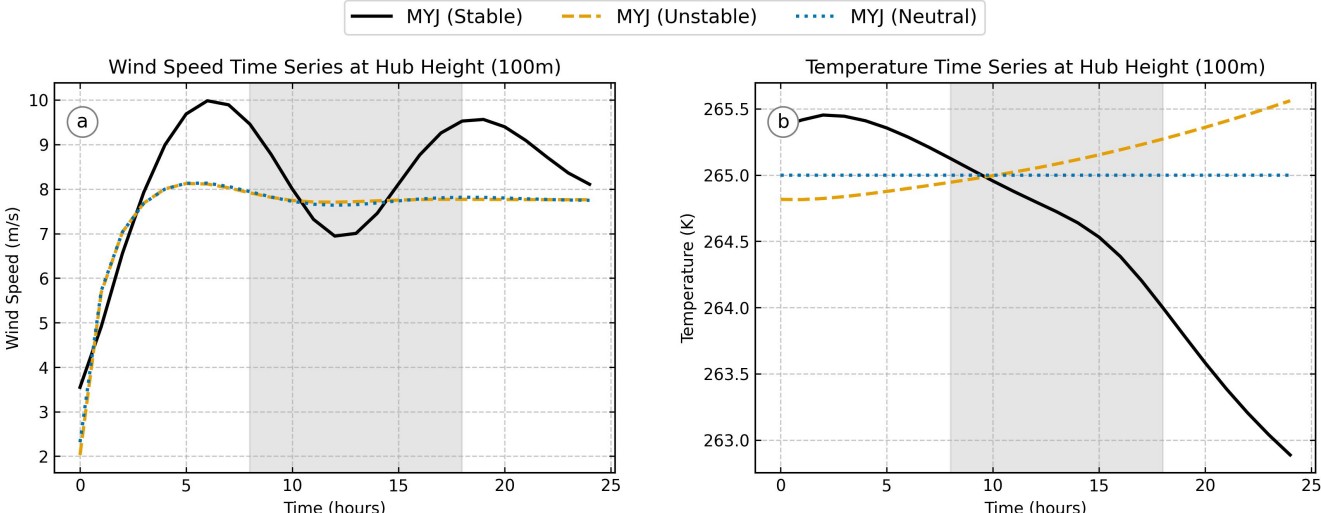

**Figure 9.** A comparison of boundary layer parameterizations under different stability and resolution. Time series of wind speed (a) and temperature (b) at hub height illustrate the differences between stable, neutral, and convective conditions on otherwise similar forecasts, indicating the frailty of assuming neutral conditions. The time series show the full 24-hour simulation, where the first 8 hours (0-8h) represent the spin-up period. The subsequent analysis focuses on the 8-18 hour window.

speed forecasts in complex terrain are highly sensitive to the choice of PBL scheme, with forecast errors larger in unstable boundary layer conditions.

The temperature deviation results (Figure 11b) further illustrate how thermal representation varies between model configurations. These temperature differences directly impact turbulence generation and suppression, thereby affecting the momentum transfer within the boundary layer that controls wind shear characteristics at hub height.

## 7 Conclusions

This study aims to evaluate the boundary layer structure, thermodynamic forcing mechanisms, and model parameterization uncertainties that affect wind conditions at turbine hub height. Our analysis reveals:

– New England experiences important seasonal changes in boundary layer structure, with hub-height to PBL ratios exceeding 2.0 during summer, indicating turbine operation above the boundary layer. Florida maintains more consistent conditions with hub-height ratios below 0.5 year-round.

– Significant diurnal PBL height variations throughout the year are observed in New England, with spring/summer differences of 400-450m between day and night in both stability regimes, while winter exhibits more moderate variations below 250m.



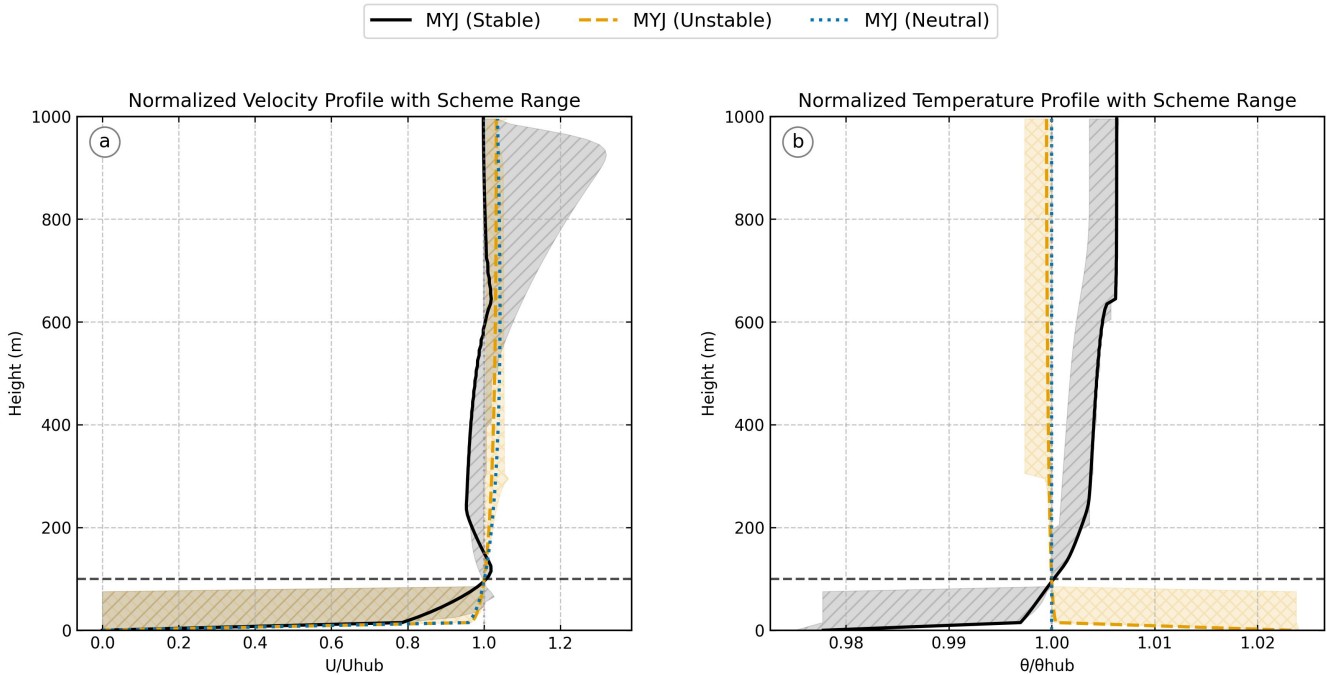

**Figure 10.** A comparison of boundary layer parameterizations under different stability and resolution. Vertical profiles of normalized velocity (a) and potential temperature (b), where shading indicates the span of results when parameterizations and resolutions are varied across those listed in Table 1.

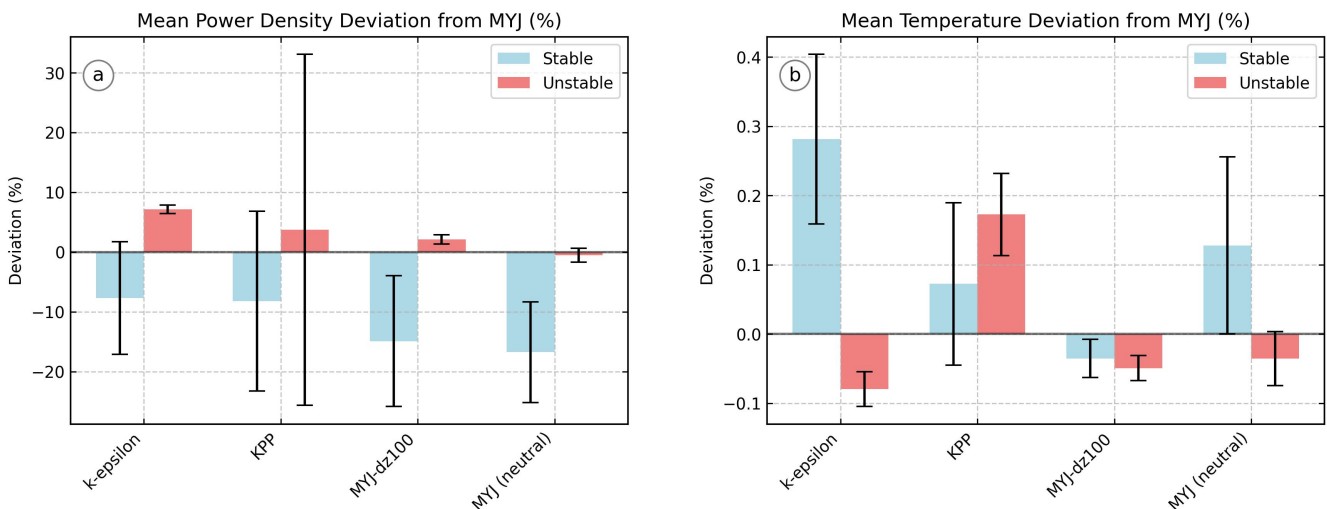

**Figure 11.** A comparison of boundary layer parameterizations under different stability and resolution. Categorized discrepancies in power density (a) and temperature (b) predictions across parameterization schemes (MYJ vs. k-epsilon, MYJ vs. KPP, all at 10m resolution), resolution (100m vs. 10m), and assuming neutral stability conditions instead of using the true conditions.



- CAPE and CIN distributions show distinct regional patterns - New England exhibits strong seasonal contrasts (winter CAPE <500 J/kg, summer >1500 J/kg), while Florida maintains consistently higher CAPE year-round.

490 – NAM parameterizations show systematic biases in turbulence representation.

- A comparison among different turbulence closure schemes produces 7-15% variations in power density estimates. Vertical resolution effects add 10-20% additional uncertainty. Assuming neutral stability when conditions are stable induces errors of 10-20% in power density estimates.

Future research should focus on reducing uncertainties. Perhaps the easiest (although computationally costly) is to increase 495 the vertical resolution of forecast systems over the PBL. High-resolution large eddy simulations carried out on extreme cases (especially stable boundary layers and wintertime diurnal cycles) can be used to study the uncertainties quantified here. Further developments in boundary layer forecast accuracy may be achieved by coupling 1D atmospheric boundary layer models with wave and ocean components to understand air-sea exchanges and their impacts on floating wind turbine performance.

*Author contributions.* **AA:** Writing – original draft, formal analysis, methodology, conceptualization, visualization; **BFK:** Writing – review 500 and editing, methodology, conceptualization, supervision, funding acquisition; **NJML:** Writing – review and editing, conceptualization, data curation.

*Competing interests.* The authors declare that they have no known competing financial interests or personal relationships that could have appeared to influence the work reported in this paper.

*Acknowledgements.* This work was supported by US Department of Energy grant DE-SC0024572. We would like to acknowledge high-505 performance computing support from the Derecho system (doi:10.5065/qx9a-pg09) provided by the NSF National Center for Atmospheric Research (NCAR), sponsored by the National Science Foundation.





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
