# Peer review of "Boundary layer structure and parameterization uncertainties affecting wind turbine hub-height predictions: A comparative assessment of New England and Florida"

_Wind Energy Science, 2025_

## Referee Comment (RC1)

**Reviewer comments on "Boundary layer structure and parameterization uncertainties affecting wind turbine hub-height predictions: A comparative assessment of New England and Florida"**

August 19, 2025

**1 General comments**

The manuscript investigates the seasonal and diurnal variation of modeled boundary layer height, atmospheric stability, and wind shear in the NAM model, and discusses how these variables depend on boundary layer parameterization and vertical model resolution. These topics are well motivated, highly relevant, and appropriate for the WES journal. Two different sites are analyzed: New England and Florida. Their contrasting meteorological conditions are clearly introduced.

The 6-hourly model output is binned according to stability and time of day. Statistics are then presented in terms of averaged temperature profiles, wind speed profiles, boundary layer height, calculated Obukhov length, CAPE, and CIN. An analysis of the resolved model TKE is also included. Finally, a one-column model is run to test different parameterizations of turbulent vertical momentum and heat transport, as well as vertical resolution.

While the manuscript addresses important and relevant topics, I find several serious issues that need to be addressed. In particular, key parts of the methodology are difficult to follow, and in several instances, the presented results differ substantially from what would be expected, without sufficient discussion or justification. My main concerns are outlined under Specific comments.

Given the number of issues identified and the broad scope of the current manuscript, it may be difficult to address all points within a revision. A more effective approach could be to focus on one central theme, where the analysis can be carried out more rigorously. In particular, concentrating on stability, boundary layer height, and wind shear would provide a solid foundation for a clear manuscript.

**2 Specific comments**

1. The diurnal variation is calculated between 0 UTC (night) and 6 UTC (day). However, local summer time in New England is UTC–4 hours and UTC–5 in Florida. Therefore, 6 UTC actually corresponds to nighttime. I recommend instead analyzing model outputs at 6 UTC for night and 18 UTC for day.

2. Fig. 31c: The potential temperature decreases with height throughout the entire boundary layer for stable conditions in both New England and Florida, which is highly unusual. Could you provide an explanation? Might this be related to averaging, to the calculation of the Obukhov length, or perhaps the figure shows temperature rather than potential temperature? Furthermore, in the text you state that $\Theta/\Theta_{hub}$ increases with height for stable conditions (Line 252), yet the figure appears to show the opposite.

3. The description of the RANS simulations in Section 3.3 is hard to follow. In Fig. 9, the results do not converge to a steady state, which is normally expected for RANS simulations. Additionally, the KPP scheme is applied to the atmosphere. Since this scheme was developed for upper ocean mixing (Large et al., 1994), its applicability to the atmosphere should be discussed. Please note that the described RANS simulations represent a simplified atmosphere, so it is expected that the results differ substantially from the NAM model. If these simulations are included, I recommend adding a clear discussion of their limitations.

4. You describe wind shear as $U/u_*$ (line 259). Could you explain the reasoning behind this definition? In general, wind shear is the change in wind speed with height. For wind energy applications, the shear exponent $\alpha$ (e.g., Draxl et al., 2014) is often used, especially because design standards (IEC, 2019) are based on it. Using such a metric would allow comparison of shear values from the NAM model to existing literature.

5. Fig. 4a: The ratio $H_{hub}/H_{pbl}$ in New England is largest under convective summer conditions, suggesting that the boundary layer height is smallest during the most convective periods. This is highly unexpected. Could you provide an explanation for this unusual behavior? In addition, the error bars in Fig. 4 show negative values, which is difficult to interpret.

6. Section 6.1: A resolved TKE is calculated as $0.5(u_*^3 + w_*^3)^{2/3}$ and compared to the parameterized TKE. It is not clear to me where this expression for resolved TKE comes from. Is $w_*$ the convective velocity scale? At a horizontal grid spacing of 12 km, microscale turbulence cannot be explicitly resolved. Without a more detailed introduction to the methodology, it was difficult to follow Section 6.1.

**3 Technical corrections**

The following comments mainly address clarity of the text, where in my opinion the presentation should be improved.

1. Lines 6–8: The sentence "In New England, hub-height to PBL ratios frequently exceed critical thresholds during summer convective conditions, indicating turbine operation above the shallow atmospheric boundary layer, while Florida maintains more consistent conditions with ratios well below these thresholds year-round." is quite dense for the Abstract. I recommend simplifying the wording, for example: "In New England, the modeled boundary layer height ranges between xx and yy, and is smaller than the hub height of modern wind turbines. In contrast, in Florida the modeled boundary layer heights remain well above turbine hub heights throughout the year."

2. Lines 25–26: "Modern wind turbines operate at hub heights of 80–140 m, a region within the upper boundary layer or its 'cap', i.e., the transition to the free atmosphere (Sanz Rodrigo et al., 2017)." Since boundary layer height varies strongly with location and stability, and is often substantially larger (500–1000 m), the statement should be weakened. For example: "Under certain conditions, boundary layer heights can be as low as or lower than turbine hub heights."

3. Lines 29–31: "Significant vertical shear and low-level jets occur during stable nighttime conditions that are frequent year-round and can affect turbine rotors, potentially beneficially." Please consider adding "often" before "occur".

4. Figures 1 and 2: Please add in the caption which model was used and over which time period the data were averaged.

5. Line 78: "and the MYJ PBL scheme used", the MYJ is introduced here for the first time, write it out here instead of in line 96.

6. Line 78: Introduce the abbreviation: Reynolds-averaged Navier-Stokes (RANS)

7. Eq. 1: The variables in the second term are introduced much later in the text, except for $S_e$. You could either note this here, introduce them earlier, or summarize them into one representative variable.

8. Lines 123–125: "This turbulence-based definition reflects a fundamental characteristic of the planetary boundary layer: the height at which turbulent mixing becomes insufficient to overcome stratification. In the atmosphere, this turbulent region is termed "mixing layer"", the connection of these two sentences is somewhat confusing.

9. Lines 128–134: The importance of the MYJ scheme could serve as a useful motivation for Section 2. I suggest moving this paragraph to the beginning of Section 2 where model parameterizations are first introduced.

10. Line 171: The sentence "These parameters sketch the potential for background conditions to form convective turbulence, which influences boundary layer characteristics." I would argue that "deep convection" is more accurate for CAPE and CIN, than "convective turbulence".

11. Lines 175–177: The definition of the Level of Free Convection (LFC) could be made clearer. The LFC is the height to which a surface parcel must be lifted before it becomes buoyant and rises freely.

12. Lines 178–179: "The Equilibrium Level (EL) marks the upper boundary where this convective mixing ceases. Between these levels, enhanced turbulent mixing is expected to affect wind turbine wake recovery and power fluctuations directly." I don't quite agree: CAPE does not necessarily lead directly to turbulent mixing. Rather, it represents the amount of energy available if air parcels rise to the LFC and overcome the CIN.

13. Section 3.3: This section is difficult to follow. If I understood correctly: the RANS model solves Eqs. 15–18, using different boundary layer schemes to obtain $K_m$ and $K_h$. Inflow conditions are specified using the wind and temperature profiles from Eqs. 19–24. Could you clarify this further? Also, please specify what is assumed for $V_g$, $U_g$, and $\Theta/\Delta t|_{rad}$ in Eqs. 15–17, and indicate the source of the stability functions in Eqs. 21–24. A short description of the KPP and $k$–$\varepsilon$ schemes (their purpose and assumptions) would also be very helpful.

14. Figures 3–5: Please state the chosen hub height either in the captions or in the main text.

15. Figure 3a–d: Please specify how the error bars were calculated (e.g., interquartile range, percentiles).

16. Figure 4: Since hub height is a constant, it is not immediately clear what is gained by plotting $H_{hub}/H_{pbl}$. The plot would be easier to interpret if absolute boundary layer height were shown directly. A horizontal line indicating typical hub heights could be added. The same applies to panels (c) and (d). Additionally, I don't understand why you are comparing the hub height to the Obukhov length. Please clarify if there is a specific reason for doing so. Note that stability classes have already been defined based on the Obukhov length. What additional information do you want to show with panels (c) and (d)?

17. Lines 284-285: "In New England, the ratio $H_{hub}/H_{pbl}$ depicts a monthly variability with a peak in July, where the mean exceeds 2 under CBL conditions, indicating that turbine hubs frequently operate above the shallow atmospheric boundary layer. This anomaly indicates that turbines may operate above the PBL." The second sentence begins with a repetition of the first. Could you shorten it?

18. Lines 347–350: "In New England (Figure 6a), limited CAPE (rarely exceeding 500 J kg$^{-1}$) and moderate CIN values are observed during winter. This environment reflects predominantly stable atmospheric conditions that characterize the northeastern United States during winter months." The second sentence does not directly follow, since you state that CIN values are small. Also, Fig. 2a shows that over 40% of winter profiles in New England were classified as unstable. It would be useful to comment on how CAPE/CIN compare to your stability classification.

19. Line 349-350: "However, in summer (Figure 6c), large CAPE values (exceeding 1500 J kg$^{-1}$) and stronger CIN are observed, suggesting favorable conditions for convection." Yet, one might argue that larger CIN values prevent deep convection, could you be more precise?

20. Line 440-443: "The time series of hub-height wind speed (Figure 9a) reveals significant differences between stable, unstable, and neutral atmospheric conditions. Under stable stratification, the simulations capture the development of a pronounced low-level jet feature, with wind speeds approaching 10 $m/s$ at certain periods." From the timeseries in Fig. 9 there is no reason to conclude that this is a low-level jet. The conclusion can only be reached using Fig. 10, where a vertical profile is given.

21. In Fig. 9, 10, the caption includes: "under different stability and resolution," yet no resolution information is given.

**References**

Draxl, C., Hahmann, A. N., Pena Diaz, A., and Giebel, G.: Evaluating winds and vertical wind shear from Weather Research and Forecasting model forecasts using seven planetary boundary layer schemes, Wind Energy, 17, 39–55, https://doi.org/10.1002/we.1555, 2014.

IEC: IEC 61400-1 Ed4: Wind turbines - Part 1: Design requirements, standard, International Electrotechnical Commission, Geneva, Switzerland, URL https://webstore.iec.ch/publication/29360#additionalinfo, 2019.

Large, W. G., McWilliams, J. C., and Doney, S. C.: Oceanic vertical mixing: A review and a model with a nonlocal boundary layer parameterization, Reviews of Geophysics, 32, 363–403, https://doi.org/10.1029/94RG01872, 1994.